# The Evolution of a Large Biobank at Mass General Brigham

**DOI:** 10.3390/jpm12081323

**Published:** 2022-08-17

**Authors:** Natalie T. Boutin, Samantha B. Schecter, Emma F. Perez, Natasha S. Tchamitchian, Xander R. Cerretani, Vivian S. Gainer, Matthew S. Lebo, Lisa M. Mahanta, Elizabeth W. Karlson, Jordan W. Smoller

**Affiliations:** 1Mass General Brigham Personalized Medicine, Boston, MA 02115, USA; 2Brigham and Women’s Hospital, Boston, MA 02115, USA; 3Mass General Brigham Research Information Science & Computing, Boston, MA 02115, USA; 4Massachusetts General Hospital, Boston, MA 02114, USA

**Keywords:** Mass General Brigham Biobank, biobank, biorepository, personalized medicine, precision medicine, translational research, research operations

## Abstract

The Mass General Brigham Biobank (formerly Partners HealthCare Biobank) is a large repository of biospecimens and data linked to extensive electronic health record data and survey data. Its objective is to support and enable translational research focused on genomic, environmental, biomarker and family history associations with disease phenotypes. The Biobank has enrolled more than 135,000 participants, generated genomic data on more than 65,000 of its participants, distributed approximately 153,000 biospecimens, and served close to 450 institutional studies with biospecimens or data. Although the Biobank has been successful, based on some measures of output, this has required substantial institutional investment. In addition, several challenges are ongoing, including: (1) developing a sustainable cost model that doesn’t rely as heavily on institutional funding; (2) integrating Biobank operations into clinical workflows; and (3) building a research resource that is diverse and promotes equity in research. Here, we describe the evolution of the Biobank and highlight key lessons learned that may inform other efforts to build biobanking efforts in health system contexts.

## 1. Introduction

The Mass General Brigham Biobank (Biobank) was founded in 2008 as an enterprise-wide initiative to drive medical discovery across Mass General Brigham (then called Partners HealthCare System). Mass General Brigham (MGB) is a large, integrated healthcare system located primarily in the Boston area that serves 1.5 million patients each year and includes several teaching hospitals, including Massachusetts General Hospital (MGH) and Brigham and Women’s Hospital (BWH). The initiative is led by Mass General Brigham Personalized Medicine, an enterprise-wide team focused on the integration of genetics and genomics into clinical care and research activities [1].

The mission of the Biobank is to enable and support translational research across MGB. Since its founding, the stated objectives of the Biobank have been twofold: (1) to accelerate the pace of discovery by providing biospecimens to investigators promptly and (2) to decrease the cost of research by providing biospecimens at a low cost thanks to economies of scale. In the first few months of its existence, Biobank leadership made the decision to focus on blood and blood derivatives rather than tissue biospecimens due to the higher cost of managing tissue biospecimens and the better alignment of blood biospecimens with the translational mission of the Biobank. 

In 2008 and 2009, a small team was formed to design the Biobank’s protocol, governance, and baseline study procedures and to implement the core systems required to run a large recruitment and biospecimen management engine. For the first few years of its existence, the Biobank relied on in-person interactions to drive recruitment, informed consent, and biospecimen collection. Biobank Research Coordinators would approach patients in the clinical setting, usually in a waiting room, and ask them whether they would be interested in learning more about the Biobank and if they would provide their consent to join the study. The consent process would happen in this outpatient setting.

Two additional recruitment mechanisms have helped augment in-person recruitment by Biobank staff. The first consists of partnerships with other studies at MGB that also recruit patients and collect blood biospecimens. These partnerships with collaborating studies have enabled the recruitment of 28,705 (21%) participants into the Biobank since 2010. A second key mechanism for recruitment is electronic consent, an automated workflow whereby patients with upcoming appointments are sent an email invitation to join the Biobank, which has enabled recruitment of 25,738 (19%) participants since 2014. They can join by logging into the Patient Portal at Mass General Brigham using their unique username and password and providing their consent electronically via a web site integrated with the Patient Portal. Their blood biospecimen is collected at a later date. From 2012 through 2020, some biospecimens were collected as discards from the clinical workflow. Since 2018, the Biobank has been placing research phlebotomy orders in the electronic health record system (Epic, Epic Systems Corporation, Verona, USA) at MGB. 

The Biobank strives to collect blood biospecimens for all consented participants. This can be challenging when participants are recruited electronically or in an outpatient setting when a blood specimen is not being collected for clinical care. Altogether, the Biobank has collected blood biospecimens for 68% of its participants.

The Biobank has obtained genomic data for about 65,000 of its participants, either by performing microarray genotyping at the Laboratory for Molecular Medicine at Mass General Brigham Personalized Medicine, or through collaboration with large research studies that are funded to obtain whole exome sequencing data and whole genome sequencing data. These genomic data have been distributed to 247 studies at Mass General Brigham.

## 2. Recruitment Mechanisms

### 2.1. In-Person Recruitment

From the Biobank cohort, 77% were recruited into the study during an in-person interaction with clinical research staff—79% of these by staff who were funded by the Biobank and 21% by staff funded by a collaborating study. Recruitment and consent interactions typically happen in clinic waiting rooms, at the periphery of clinical workflows. In order to gain access to outpatient clinics, Biobank staff engage with clinical leadership and clinical staff to formally request permission to approach patients and to be present in the clinical space. Most clinical teams are generally welcoming of these research activities. It is understood by MGB staff that research is a core pillar of the organization’s mission. 

Whenever possible, Biobank staff strive to seamlessly integrate into clinical workflows. An example of successful integration of research workflows into clinical workflows is a partnership the Biobank established with phlebotomy teams in several outpatient and hospital central phlebotomy locations, where the Biobank has successfully integrated staff certified to perform clinical phlebotomy into the phlebotomy teams. They ask all patients coming through the outpatient phlebotomy locations whether they would like to join the Biobank and perform the consent process when appropriate. They then collect blood for both clinical and research purposes. For patients who decline Biobank participation, only the clinical phlebotomy collection is performed. This model has been very successful because it benefits both the Biobank and the phlebotomy teams. However, it can be challenging to implement. Some key challenges are: (1) limited opportunity to train and certify staff for clinical phlebotomy; (2) high demand in phlebotomy locations sometimes makes Biobank recruitment infeasible; (3) lack of buy-in from phlebotomy leadership at some sites; and (4) inability to scale this model due to the complexity of our organization, and unique clinical workflows at each phlebotomy location.

For the most part, Biobank staff recruit patients without being embedded into clinical workflows. This creates inefficiencies that impact the ability to recruit participants and collect biospecimens. For example, it is not uncommon for Biobank staff to start a conversation with a patient in the waiting room only for the conversation to be interrupted by the patient being called into the exam room. It is rarely possible for the Biobank Research Coordinator to reconnect with the patient after their clinical visit. In-person recruitment was paused during the COVID-19 pandemic, from mid-March 2020 through June 2021, and then again in December 2021 through February 2022. Since resuming in-person Biobank activities, in-person recruitment has proven to be more challenging as clinical offices have implemented mitigation procedures that further limit a patient’s wait times in the waiting rooms. In addition, Mass General Brigham hosts thousands of research studies, and the Biobank frequently competes with other studies to recruit patients in the same clinics. Maintaining recruitment operations requires ongoing buy-in and support from clinical leaderships in those clinics.

### 2.2. Electronic Informed Consent

In 2014, the Biobank launched an electronic consent workflow to improve recruitment rates. According to this new workflow, patients with upcoming appointments are sent an email invitation to join the Biobank. The email is only sent to patients with accounts in our institutional Patient Portal. Patients are asked to log into the Patient Portal to learn more about the study. Once logged into the Patient Portal, they can view more information about the Biobank as well as review the consent form and associated fact sheet. Patients are thus able to provide consent electronically, without any interaction with a Research Coordinator. This mechanism is available in both English and Spanish. It has been shown that participants who enroll through a self-consent process still maintain a reasonably high level of knowledge and recall of the consent form they agreed to [2].

Since its launch eight years ago, the electronic consent mechanism has yielded a rate of consent of 2.1% among 1.2 million unique contacted patients. In addition, biospecimen collection is asynchronous and a substantial proportion of those who consent have not completed the Biobank blood draw. Despite this seemingly low yield, we believe this mechanism has been cost-effective as it requires minimal effort on the part of staff. To date, electronic consent has yielded 25,738 participants, with 55% completing biospecimen collection. One limitation, however, has been that the cohort recruited though electronic consent has been demographically less diverse than the overall Biobank (Table 1). We discuss efforts to diversify the Biobank cohort in a later section (see “Diversity and Equity”).

### 2.3. Collaborating Studies

Another successful mechanism for Biobank recruitment is the collaboration with other studies that have aligned objectives. To date, the Biobank has collaborated with 33 studies to recruit participants, yielding 28,705 enrollees to the Biobank. In this model, Biobank staff partner with the collaborating study to design a merged consent form that covers the terms of both studies. This type of partnership has worked particularly well when the collaborating study is building a biorepository focused on a specific phenotype (as opposed to the Biobank, which is not focused on any specific disease area). In such cases, the collaborating study can benefit from the Biobank’s physical and information technology (IT) infrastructure. This can help them to build up a repository of biospecimens and data faster and at a lower cost than if they were to do this on their own. They also benefit from the Biobank’s robust infrastructure as well as the generation of genomic data that they can then obtain at no cost. Figure 1 shows the contributions of the three primary recruitment methods to the growth of the Biobank over time. 

## 3. Informed Consent Form

The Biobank consent form has followed best practices for research biobanking [3]. Key elements of the consent form and accompanying “fact sheet” are shown in Table 2. Over the course of its history, the Biobank has had two significant changes to its informed consent form. The first, in 2013, was the addition of a section on return of genetic research results. New language informed participants that they may be contacted if research results revealed were “of high medical importance”. This was anticipated to primarily include genetic variants recommended by the American College of Genetics and Genomics for evaluation and reporting of actionable secondary findings [4]. The new consent form language also made clear that any research results the Biobank may find are not the same as clinical results, that it is possible participants may never be contacted with important results, and that participants should not conclude that they will not develop a health problem just because they are not contacted.

The next set of significant changes to the Biobank’s consent form occurred in 2020, and again focused primarily on return of genetic results. Where before only medically actionable single gene variants were returned, the new consent form expanded the range of results that might be returned to include polygenic risk scores, pharmacogenetic information, ancestry information, and carrier status for recessive or X-linked conditions. Because of the potentially immediate clinical relevance of pharmacogenetic results, participants are informed that these results may be placed directly into their medical record, when clinically validated assays were conducted on clinically derived biospecimens, even if the Biobank is unable to contact them. For all other results, participants can opt out of knowing their specific findings after they are contacted. As the processes for returning these results are still in development, the consent form makes it clear that it may be months or years before these results are returned, in addition to the prior language indicating that not everyone will receive results. 

In 2013 and again in 2020, the changes to the consent form were significant enough that efforts were made to contact all participants to ask them to sign a new consent form. For the 2013 reconsent effort, which was conducted via mail only, about 3% of the 3737 participants requiring reconsent signed the new consent form by 2014 (by 2022, 11% have reconsented, though via incidental recruitment by collaborating studies). For the 2020 reconsent effort, of 120,000 participants: 92,000 were emailed, 20,000 were sent a paper letter, and 8000 were ineligible for reconsent via email/post (deceased, required healthcare proxy, etc.). 17,000 of these additionally received one or more follow-up phone calls. Of those eligible to be reconsented, 10% reconsented (of which, 91% were due to the efforts of Biobank staff and 9% due to incidental reconsent via collaborator), 1% refused to reconsent, and 1% withdrew their participation entirely.

## 4. Biospecimen Collection Mechanisms

Blood specimens collected by the Biobank provide material for a range of research uses. DNA extracted from blood biospecimens and the buffy coat fraction can be used for DNA genotyping and sequencing, epigenetic/epigenomic analysis, and other uses (e.g., somatic mutation and telomere length assays). Serum and plasma can be used for a range of biomarker assays including proteomic, metabolomic, and various blood biomarker assays. The Biobank has leveraged several biospecimen collection mechanisms throughout its history, though challenges remain. During the Biobank’s first few years, Biobank Research Coordinators were trained to perform research phlebotomy and would conduct the blood draw immediately after the consent interaction. Some staff were clinically trained as phlebotomists and integrated at phlebotomy sites. Using these mechanisms, the Biobank was able to collect blood biospecimens for 92% of its first 10,000 participants from 2010 through 2013. However, as the Biobank expanded recruitment, collection of the blood biospecimen at time of consent was not always feasible and other methods of collection were explored in the following years. As of 2022, the Biobank has collected biospecimens from 68% of its participants.

The Biobank also relied on a software system to collect residual biospecimens from clinical blood draws (“clinical discards”) at BWH starting in 2012 (though discard collections did not approach meaningful collection volume until 2014), and then at MGH starting in 2015. This mechanism enabled the collection of biospecimens for 25,000 consented participants over the nine years it was active. The drawback of this method is that it allows only for extracted DNA as a specimen and is not suitable for deriving serum or plasma because of delays in processing of discarded biospecimens. Moreover, the quality and quantity of DNA derived from discard biospecimens is greatly variable due to the unpredictable volume remaining post clinical workflows and due to the delay in processing post collection.

The Biobank worked closely with pathology Information Technology (IT) leadership for several years to design a blood draw order in Epic, the electronic health record system at MGB. According to this mechanism, Biobank staff place orders for consented participants and the blood is collected at a later date whenever the participant comes in for clinical blood draws through outpatient phlebotomy. The Biobank draw consists of 30 mL of blood collected in two 10 mL purple top tube (EDTA) and one 10mL red top tube (serum tube). The ability to enter Epic orders went live in 2018 and, by the end of 2019, 28,000 orders had been placed. This process requires periodic validation that blood orders are deleted when a participant withdraws from the Biobank, when a blood biospecimen is collected through another mechanism, or when an order is released by a clinician or phlebotomist by accident without drawing blood. Figure 2 shows the numbers of samples per year attributable to each biospecimen collection method. 

With the start of the COVID-19 pandemic in March of 2020, Biobank recruitment staff ceased onsite operations and moved to remote work. This allowed the team to place thousands of Epic orders for participants for whom we did not yet have a biospecimen nor an Epic order. The Biobank team placed 24,000 additional orders in 2020 and 2021. However, it was not possible to monitor the success of this strategy due to the continued pandemic, which significantly reduced the volume of patients at our hospitals.

By March 2022, it had become clear that the mechanism to collect blood biospecimens via Epic order was providing poor results. Biobank orders were released and drawn by outpatient phlebotomists during only a fraction of clinical phlebotomy encounters with participants who had a Biobank order in Epic. For example, Biobank participants with active orders have had 8723 clinical phlebotomy encounters in 2022 so far, but only 569 of those resulted in a Biobank collection (7%). Upon consultation with phlebotomy leadership across MGB, issues such as lack of training of new phlebotomists and refusal of participants have impacted order release and blood draw rates. Given the low release rate, an order may take dozens of encounters over several years before yielding a Biobank draw.

To enhance the yield of biospecimen collection, the Biobank is designing an engagement and outreach initiative to train all outpatient phlebotomists on the importance of releasing the Epic order. This will require meeting with phlebotomists at multiple outpatient sites, on a regular basis, to inform them about the Biobank, its value and purpose, and the mechanism for releasing the Biobank order. Additionally, phlebotomists will learn about their impact on Biobank operations; biospecimens are essential to the services the Biobank provides to investigators. The Biobank also plans an email/letter campaign to participants without biospecimens to inform them about the Epic order/biospecimen collection process.

## 5. Storage Facility

The Biobank storage facility is located in Cambridge, MA, at the mid-way point between BWH and MGH. The facility currently holds more than 1.2 million specimens, with the capacity to hold up to 3 million specimens. Thanks to economies of scale, the Biobank has been able to stand up more robust physical inventory procedures than many smaller biorepositories.

To ensure specimen integrity, each freezer is internally equipped with a heated gasket that provides four touchpoints of security and seven zones of protection and are all on emergency back-up power. Each individual freezer is also monitored 24 h a day/7 days per week. This system is triggered when there is a loss of normal power, a rise in temperature within the freezer, or there is a loss of communication with the freezer’s alarm circuit. Both an email and phone call are placed to a pre-authorized list of individuals as well as to the building management security team that is on-site 24 h a day.

The storage space has two sources of cooling, an independent cooling (HVAC) system that provides 12 tons of cooling, and the existing house VAV system that provides ~28,000 CFM of 55-degree air or ~7 nominal tons of cooling. The overlay system (the independent HVAC) is designed to function automatically upon loss of normal power using an automatic transfer switch connected to the house standby generator. The automatic transfer switch is tested annually.

## 6. Biobanking Information Technology (IT) Infrastructure

MGB has made a major investment in IT infrastructure for the Biobank that enables recruitment, biospecimen management, user queries of data and biospecimen requests, and other operational functions. The Biobank has a dedicated IT team that has implemented and continuously enhances a suite of applications. The core components of this infrastructure are (1) consent tracking and recruitment workflow management, (2) biospecimen management and processing, and (3) a query tool for investigators to identify participants using demographic, electronic health record (EHR) data, and genomic data. The Biobank’s IT infrastructure is integrated with the clinical systems at MGB, including the Enterprise Master Patient Index, the Electronic Health Record (Epic), and the Patient Portal (Patient Gateway, which is built on MyChart).

Consent tracking and recruitment workflow management are performed via a custom application called CONSTRACK and an electronic consent website that is integrated with the MGB Patient Portal. Some key functionality available in CONSTRACK is the build-out of custom patient lists based on clinical scheduling data, bulk messaging through the Patient Portal, and the tracking of all interactions with participants, including capturing key metadata such as the interaction mechanism (e.g., email, paper letter, outpatient interaction, etc.), the location of the interaction (such as the clinic name) and the name of the staff person involved in the interaction. This type of data is essential for understanding and optimizing a large recruitment operation. The same infrastructure is leveraged for several other large studies at Mass General Brigham, including the NIH *All of Us* Research Program and the NHGRI-funded eMERGE Clinical Center at MGB. 

In addition, a customer relationship management application (Salesforce, San Francisco, USA) manages participant emails and phone calls as well as investigator queries and requests. A secure electronic data capture application (REDCap, Vanderbilt University, Nashville, USA) manages the Biobank health information survey.

Biospecimen management and biospecimen processing are performed through a laboratory information management system (LIMS) (STARLIMS, Hollywood, USA). It provides all specimen-handling workflows, such as inventory management, serum and plasma isolation, DNA extraction, and DNA plating. As a next step in the evolution of the LIMS, Genotyping and Next-Generation-Sequencing (NGS) functionality will be built into it. 

Investigators at MGB may search the inventory of biospecimens and a wide range of disparate data in the Biobank Portal (Figure 3); and may request biospecimens and genomic data and create and download custom data sets for analysis. This self-service, web-based tool is a flagship component of the Biobank that provides access to data from the Electronic Health Record (EHR) that are integrated and harmonized with narrative notes, biospecimen data, quantitative imaging data, and genomic data for the consented Biobank participants, and made available for querying and download. 

The Biobank Portal can return either limited data sets or identified data sets. Because Biobank participants have already consented to broad use of their data and samples, additional IRB approval is not required to obtain a limited data set through the Biobank Portal but is required to obtain identified data. Only studies with an IRB protocol that allows for the use of identified data may obtain identified data sets. In all cases, Biobank Portal users must sign a data use agreement.

The Biobank Portal also provides access to 100 computed validated phenotypes derived from the coded EHR data and natural language processing applied to the clinical notes. The Charlson Comorbidity Index is used to estimate 10-year survival probability categories for all Biobank participants based on EHR variables. In a query, a researcher can choose categories of high 10-year survival probability to select healthy controls based on this calculated index. The Biobank Portal is an open-source web application based on the Informatics for Integrating Biology and the Bedside (i2b2, TranSMART Foundation, Boston, USA) software and data model. I2b2 enables the integration of high-dimensional data types in the Biobank Portal so that investigators may construct complex queries and extract data for analytics. Data types include genomic data, which may be queried in the Biobank Portal by gene or variant; and may be requested by investigators through the tool. A wiki provides documentation and instruction, and a user liaison provides support to investigators as needed. Using the Biobank Portal requires little to no training, though users must be registered, sign an online data use agreement, and be within the institutional firewall to use it. Researchers who are not data scientists or epidemiologists, as well as those who are, are able to use the Biobank Portal to gather and extract data for feasibility studies and can easily request biospecimens and genomic data. The Biobank Portal currently has 1129 registered users.

In addition, the Biobank implemented Salesforce to manage some patient queries and to track all interactions with investigators as well as key data associated with their requests. The Biobank Portal provides self-service tools for investigators to build cohorts and request biospecimens/data, and Salesforce manages any further conversations required to close out a distribution of biospecimens or data.

Biospecimen requests are routed to STARLIMS for fulfillment. Biobank staff is responsible for managing the distribution of the biospecimens to investigators and for ensuring that data integrity is maintained through the full biospecimen lifecycle.

## 7. Genomic Data Generation and Distribution

To date, the Mass General Brigham Biobank has generated and provided genomic data for ~65,000 of its participants. The data include genotyping data performed on the Illumina Multi-Ethnic Genotyping (MEG) Array for ~36,000 participants and both genotyping data on the Illumina Global Screening Array (GSA) and exome sequencing data performed for ~54,000 participants. The generation of the GSA and exome sequencing data was conducted at the Broad Institute through funding from both the National Institutes of Health and a collaboration with the Broad Institute’s IBM Cardiovascular Disease Collaboration. All genotyped biospecimens have at least a 99% call rate per array and are checked for sex concordance using sex in the EHR and sex computed from the array data. Exome sequencing data typically meet or exceed coverage metrics, and biospecimens are not used in variant calling if they have less than 40% target bases at 20x or greater than 0.1 contamination. In addition to the variant calls from the array and exome sequencing datasets, imputed variant data derived from the arrays are available using three reference populations: Haplotype Reference Consortium, 1000 Genomes Phase 3, and TOPMed. Additional datasets for imputed HLA haplotypes and genome sequencing data are being generated for future release. In addition, investigators who use Biobank samples to generate genomic data agree (in the data use agreement) to return these data to the Biobank for inclusion into Biobank genomic data resources available to the broader MGB research community. The Biobank distributes genomic data to MGB investigators at no charge. Requests are made via the Biobank Portal and data are provided 24–72 h after the request is placed.

## 8. Biobank Services

The Biobank provides two primary services: the distribution of biospecimens (DNA, plasma, serum, and PBMCs for a COVID-19 cohort that includes acute and post-acute COVID patients) and the distribution of genomic data. Additional services include the distribution of survey data, biospecimen inventory management for private studies, biospecimen processing services, and recontact of Biobank participants to invite them to participate in other studies. Through its affiliation with the Biobank Genomics Core and the Laboratory of Molecular Medicine within the confines of MGB, the Biobank also provides genotyping and sequencing services, including data generation in a CLIA-certified laboratory.

A key component of the value that the Mass General Brigham Biobank provides to its institutional research community is the linkage of the biospecimens and genomic data to the electronic health record [5]. The Biobank has distributed close to 153,000 biospecimens to 208 studies and has distributed genomic data to 247 studies led by MGB investigators. In addition, the Biobank has recontacted 57,255 participants to ask them to participate in another study. The rate of consent for this recontact service varies significantly from study to study. 

To perform these distributions as efficiently as possible, the Biobank has implemented a customer relationship management system using Salesforce software to manage all investigator inquiries and requests. This system enables the Biobank to track all communications with investigators in a centralized location, to leverage pre-written responses to address common questions, and to systemically structure and organize all inquiries as well as queries.

## 9. Biobank Governance and Oversight

The Biobank is led by an executive committee comprising senior investigators from participating hospitals (MGH, BWH, Mass Eye and Ear, and McLean Hospital). Organizationally, the Biobank is in Mass General Brigham Personalized Medicine and reports into the Chief Academic Officer’s Office.

The MGB Institutional Review Board (IRB) reviews all Biobank study procedures and meets with Biobank leadership monthly to discuss planned changes to the protocol as well as any unusual issues that have been encountered. Due to the scale of the Biobank, this monthly meeting with IRB leadership has been invaluable over the years. 

The Biobank distributes biospecimens and data only to investigators who are formally affiliated with an MGB institution. It is not possible for external entities to request biospecimens or data directly, but it is possible for external entities to obtain them through a data use agreement or material transfer agreement in the context of a collaboration with an MGB investigator. The Biobank’s biospecimen distributions are high-touch, with a Biobank team member overseeing the process from beginning to end. This includes ensuring the investigators have executed proper data use and material transfer agreements and checking that appropriate IRB approvals are in place. All research issues are reported to the IRB and, if they concern a privacy issue, to the institutional privacy offices. All biospecimen and data distributions are reviewed by the Biobank leadership team on a monthly basis.

## 10. Biobank Cost Model

As a research core and according to the terms of its consent form, the Biobank does not return a profit on any of the services it provides to investigators. In 2012, when it performed its first biospecimen distribution, the Biobank established a fee-for-service cost structure in order to cover operational expenses, including recruitment, biospecimen collection, processing, and distribution [6]. The cost of biospecimens and processing services are designed to cover the operational costs of running the Biobank, including recruitment, biospecimen management, IT infrastructure, and biospecimen plus data distribution. Ideally, all Biobank operational costs would be recovered via this fee-for-service cost structure. However, biospecimen fees received to date have never fully covered operational costs. As such, the Biobank remains partially funded at the institutional level.

Financial sustainability is a common challenge in biobanking and implementing a cost recovery model is vital to ensuring the future of the Biobank [7]. The costs of running a large Biobank operation are significant, and it is a challenge to design the most cost-effective mechanisms to sustain such an operation [8]. The National Cancer Institute Biorepositories and Biospecimen Research Branch conducted a survey of biobank managers and directors to learn more about the economic considerations of biobanking. Of 106 responses, 27% of respondents reported no recovery of operational costs and 42% reported that they recover 1–25% of their costs via a cost recovery model [9].

To facilitate the use of Biobank data by the MGB research community, the Biobank distributes genomic and survey data to MGB investigators without charge. There is no study set-up fee given that data requests are made via online tools, but there is a fee associated with the storage of the genomic data, which is covered by a distinct research core and not by the Biobank. The return of investment based on grants received vs. institutional funds invested has been about 10:1.

Some biobanks have benefited from integrating their recruitment structures into clinical workflows. Others have established collaborations with industry partners, such as pharmaceutical companies, to cover the cost of key components, such as the generation of genomic data. For example, Geisinger entered into a partnership with Regeneron to generate genotype and exome sequence data on its participants [10]. Similarly, the MGB Biobank relied on a partnership with IBM to support the generation of exome sequence data.

The Biobank continues to evolve efforts to achieve a higher degree of sustainability and limit the reliance on institutional funds. One approach has been to expand the number of collaborating studies across the institution to leverage existing studies that are collecting biospecimens. Another mechanism would be to seamlessly integrate recruitment workflows into clinical workflows. This is a challenge in an institution that hosts thousands of parallel research studies, all focused on approaching patients to ask them to participate in a separate study.

## 11. Return of Research Results

Offering return of results (RoR) to participants in a research setting, once ethically controversial, is becoming standard practice [11]. For example, in 2021 the Global Alliance for Genomics and Health approved a policy for the return of clinically actionable genomic research results to research participants. Providing actionable genetic results provides an important opportunity to provide value to participants. At the same time, returning results to Biobank participants straddles the line of research and clinical care. To ensure a successful return process, appropriate language in the consent form discussing the possibility of results, a clear protocol, and genetic specialists such as genetic counselors are needed at a minimum [12].

The MGB Biobank has been using research results to identify participants with potential actionable findings since 2017. The consent form explicitly states that participant’s DNA will be analyzed for research, that they might be recontacted if “medically important” results are discovered, and that enrollment in the Biobank entails consent for this recontact. Research genomic data are reviewed to identify likely pathogenic/pathogenic (LP/P) variants in a set of actionable genes, currently compromising version 3.0 of the American College of Medical Genetics and Genomics secondary findings list (ACMG SF v3.0) [13] plus additional genes that were considered actionable by the MGB Biobank Return of Results (RoR) Committee.

Details of the RoR process have been described elsewhere [14]. In brief, we use an “incremental disclosure protocol” beginning with the informed consent process where participants can opt out of receiving their research result during multiple phases. Living participants are eligible to receive results if their finding is not already documented in their medical record. As currently these variants are identified from research data, results are not specifically shared with the participant nor put into the medical record when identified. Instead, the patients are contacted for consent, and a biospecimen is collected for clinical confirmation of the research finding. A Biobank study genetic counselor (sGC) contacts eligible patients via mail and phone to share the research finding with the participant, discuss details, benefits, and risks of confirming the finding, and to facilitate decision making. If a participant decides to pursue clinical confirmation, a saliva kit is mailed to the participant’s home. The saliva kit is mailed back to the laboratory and undergoes targeted Sanger sequencing for the identified variant.

As the clinical laboratory is processing the biospecimen, the sGC facilitates scheduling for a disclosure appointment with a provider (a medical geneticist, disease specialist, or the participant’s own primary care physician if requested). This appointment is a conventional clinical appointment to provide contextualization of the result, discussion of further screening or preventive strategies, and facilitation of appropriate medical follow-up. The responsibility of the research team is considered to end with this disclosure appointment and the result is documented in the clinical record. As referenced above, the participant can opt out after the initial phone call with the sGC, after receiving the saliva kit in the mail, and/or before scheduling an appointment to discuss their result with a provider.

In our detailed analysis of the RoR process during 2017–2021, we found that more than 75% of participants who carried actionable variants were previously unaware of this, underscoring the added value of genetic RoR in this research setting [14]. Of note, 37% of those contacted regarding actionable results opted out of further disclosure. In an economic analysis, we estimated that the cost of the RoR protocol (beyond the initial research genotyping or sequencing costs) averaged $14 per participant overall and $3224 for each participant for whom gRoR was successfully completed.

## 12. Evolution of the RoR Results Process

Since the initiation of the MGB Biobank RoR process, we have implemented necessary changes to keep up with the evolution of science and policy. For example, The Interoperability, Information Blocking and ONC Health IT Certification Final Rule was published in May 2020 as part of the 21st Century Cures Act. Under this rule, electronically available health results must be disclosed to patients as soon as they become available. Previously, the Biobank clinical genetic report was not released to the patient’s EHR until the patient met with a genetics provider for the disclosure appointment. In light of the information blocking rules, the clinical report is now made available to patients as soon as the laboratory testing is completed and the report has been generated. In the current workflow, once the clinical result is available, a participant can no longer opt out of the result being put into their medical record unless specified during the pre-test counseling phone call. They can now, however, opt out of meeting with a provider to discuss their result.

As noted earlier, in 2020 the Biobank modified its consent form in anticipation of future expansion of RoR to include polygenic risk scores, pharmacogenetic results, and ancestry. These result types have not yet been generated for participants and workflows to enable such returns are under development. It is anticipated that these results would be returned via primary care providers with clinical decision-support information developed by the Biobank.

## 13. Measuring the Value of the Biobank

When the Biobank was founded, advances in genomics and bioinformatics increased emphasis on translational research, which in turn stimulated the demand for stored specimens and associated data. At the time, few non-profit Biobanks reported that they experienced a great deal or a moderate amount of competition [15]. With the growing availability of large-scale volunteer research repositories such as the UK Biobank and the *All of Us* Research Program, a key question has emerged: what is the value of institutional Biobanks and how much should institutions invest in these?

It has been suggested that the sustainability of Biobanks will require a detailed analysis of the inputs and outputs of the operation to clearly demonstrate their value [16]. Considering this, the Biobank has been tracking a variety of metrics including institutional investment, investigator funding related to use of Biobank resources, publications utilizing Biobank data or samples, and other measures of client utilization (Table 3, Table 4, Table 5 and Table 6). The MGB Biobank leadership team expects that our Biobank will continue to enable and drive medical research across the institution. Advantages of a local biobank relative to other large-scale national repositories is the ability to provide researchers with the opportunity to access identified EHR data for phenotype validation and recontact of participants (including by genotype) for investigator-initiated studies, as well as a more rapid return of genetic results to participants to promote the clinical management of unrecognized risk. 

The scientific dividends of the Biobank have already been substantial. For example, MGB Biobank data have been used in studies demonstrating the limitations of contemporary guidelines for managing patients at high genetic risk of coronary artery disease, to evaluate polygenetic Risk Score models for prostate cancer, and to identify extensive adrenal suppression due to inhaled coriscosteroid therapy in asthma [17,18,19]. It also provides the foundation to manage large NIH-funded research programs that require IT support that can scale and staff with large-scale recruitment and biobanking experience. 

## 14. Diversity and Equity

At its inception in 2008, Biobank operations were designed to emphasize the rapid scaling of the Biobank resource by recruiting large numbers of participants. As a result of this focus on scale, the Biobank established its core operations at the two largest hospitals at Mass General Brigham: MGH and BWH. Recruitment operations were established in clinics, including phlebotomy sites, that see the largest number of patients. This focus on recruiting patients at high-volume clinics at the main campuses of MGB’s largest hospital drove the demographic makeup of the Biobank. However, reflecting the patient populations at these recruitment sites, the Biobank cohort is primarily comprised of self-reported White participants, with only 12% of Biobank participants self-identifying as Black, Hispanic/Latino or Asian.

In 2014, the Biobank began recruitment at community health centers associated with MGH and BWH, improving the diversity of the cohort (Table 6). For example, between 2015 and 2017, 1015 participants were recruited at Chelsea Community Health Center. This cohort is primarily Hispanic/Latino, reflecting the community at Chelsea. These gains in diversity were partially offset with the introduction of electronic informed consent procedures in 2014. Participants recruited through this mechanism were less diverse, likely due to the “digital divide” [20]. Since 2017, MGB has also been a clinical center in the *All of Us* Research Program, which prioritizes recruitment of communities that have been historically under-represented in biomedical research. *All of Us* program recruitment at the community health centers has sometimes limited Biobank activities in these more diverse settings.

In recent years, the importance of improving diversity in genomic and precision medicine research has been recognized as an urgent priority [21,22,23]. It has also become a top priority for the MGB Biobank. Lack of diversity in research cohorts can mean that scientific advances do not equally benefit all populations, exacerbating health disparities and undermining equity in healthcare. To address this, the MGB Biobank is launching an initiative to increase the diversity of its participants. This will include increasing recruitment efforts at community health centers, partnerships with institutional programs focused on diversity, equity and inclusion, and shifting the Biobank’s focus onto community engagement initiatives. 

## 15. Conclusions

The MGB Biobank has evolved to become an invaluable resource that has helped drive biomedical discovery and the translation of these discoveries into clinical care. In the process, it has helped to pioneer innovations that are now more widely adopted including electronic informed consent, IT infrastucture linking EHR and biospecimens, and return of genetic results. At the same time, the growth and maintenance of the MGB Biobank has required substantial institutional investment. Although the return on this investment has clearly been high, its sustainability remains an ongoing challenge and will necessitate new efforts at enhancing revenue and recovering costs. Table 7 summarizes some of the key lessons learned over the lifetime of the MGB Biobank. We hope that these lessons can be of value in informing similar efforts in other health systems. The Biobank has been a dynamic entity that will continue to adapt to the needs of the research and clinical communities we serve. 

## Figures and Tables

**Figure 1 jpm-12-01323-f001:**
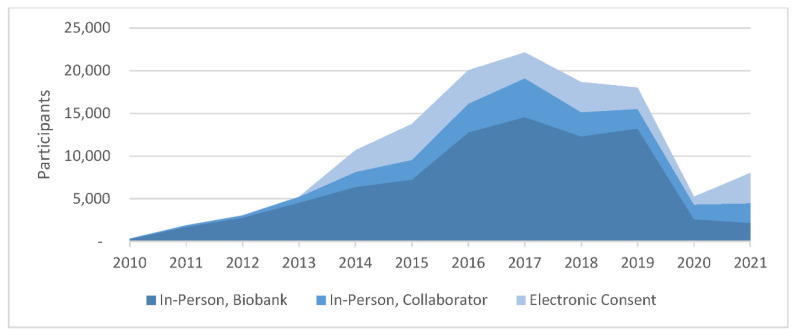
Consent by Recruitment Method.

**Figure 2 jpm-12-01323-f002:**
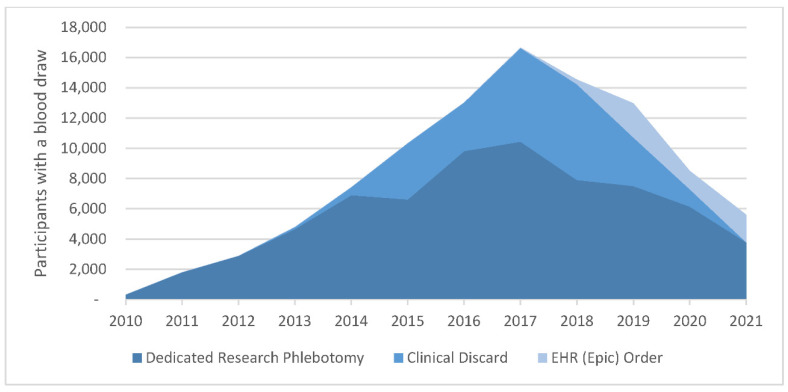
Biospecimen Collection by Method.

**Figure 3 jpm-12-01323-f003:**
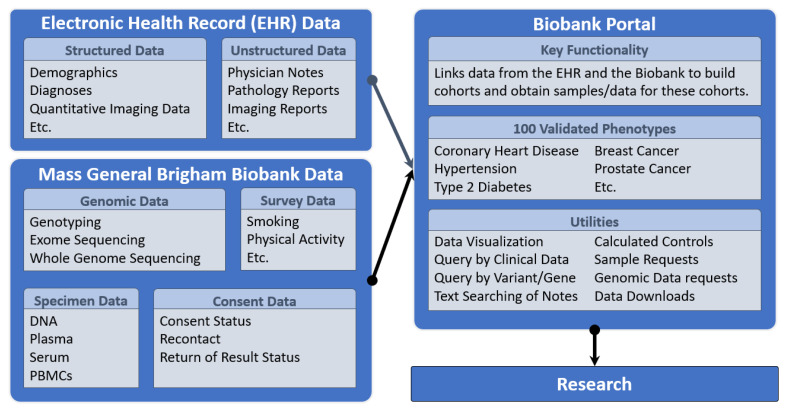
Biobank Portal Data Integration for Automated Access to Clinical Data.

**Table 1 jpm-12-01323-t001:** Demographics of Electronic Informed Consent Compared to Demographics of the Full Biobank.

Sex	Electronic Consent	In-Person Consent
Female	63%	55%
Male	37%	45%
Race/Ethnicity	Electronic Consent	In-person Consent
American Indian/Alaska Native	0.1%	0.1%
Asian	3.9%	2.6%
Black	1.4%	5.2%
Hispanic	2.8%	5.9%
Other	0.9%	1.0%
Two or more	0.8%	0.6%
Unknown	2.6%	2.5%
White	87.4%	82.1%
Age	Electronic Consent	In-person Consent
20–39	24%	21%
40–59	29%	28%
60–79	41%	41%
80–99	6%	10%
Education	Electronic Consent	In-person Consent
8th Grade or Less	0%	1%
Some High School	0%	2%
High School/GED	7%	18%
Some College	6%	10%
College	50%	42%
Post Grad	21%	13%
Other	3%	4%
Unknown	12%	10%

**Table 2 jpm-12-01323-t002:** Key elements of MGB Biobank informed consent form and fact sheet.

Consent Section	Consent Form Topic	Fact Sheet Topic
Purpose	Study how genes and other factors contribute to disease	Yes
Procedures	Fresh blood biospecimen (up to 5 tubes) and future discarded specimens (blood, urine, tissue)	Yes
	A possible future biospecimen of up to 3 tubes	Yes
	Biospecimens linked to electronic health record	Yes
	Questionnaires about health behaviors and family history	Yes
	Re-contact for other information or studies	No
Research Conducted	Biological and genetic research	Yes
	DNA analysis and how changes in DNA affect health	Yes
Return of Results	Return of results is unlikely, but participants may receive research results of high medical importance	Yes
	Contacting participants and/or their providers with results and/or placing results in a participant’s medical record	Yes
	Single gene variants	No
	Polygenic risk scores	No
	Pharmacogenetics/how genes influence response to medication	No
	Results unrelated to health that may be of interest to participants	Yes
	Research results are not the same as clinical tests	Yes
	Patient and insurer may be responsible for costs of tests and follow-up care	No
Benefits	May not directly benefit, but may help us understand, prevent, treat, or cure disease	Yes
	May directly benefit if researchers find results that are important to a participant’s health	Yes
	There is no payment for biospecimens	No
Biospecimen and Information Storage	Biospecimens are de-identified and the key to decode is stored securely	Yes
	Biospecimens and data are stored indefinitely	No
Researchers with Access	Mass General Brigham investigators	Yes
	Researchers at non-Mass General Brigham institutions must work with MGB investigators to obtain access to de-identified biospecimens	Yes
	For-profit companies must work with Mass General Brigham investigators to obtain access to de-identified biospecimens	Yes
	Central biobanks who may share coded biospecimens and data with other researchers	No
	Biospecimens will not be sold for profit	No
Withdrawing	Participants can withdraw at any time, but it is not possible to destroy biospecimens and information already given to researchers	No
Risks	Potential loss of privacy	Yes
	Influence on insurance companies and/or employers	No
	Cannot predict how genetic information will be used in the future	Yes
	Bruising or infection from blood draw	No
Certificate of Confidentiality	Researchers cannot be forced to disclose identifying information, even by a court subpoena	Yes
	Does not prevent patient from voluntarily releasing information about self-involvement in research	Yes
	Certificate does not prevent researchers from disclosing information without consent such as child abuse and intent to harm self or others	No

**Table 3 jpm-12-01323-t003:** Biobank Inputs (June 2022).

Team	9 research coordinators (the team has included up to 20 research coordinators in the past), 2 managers, 4 laboratory technicians
Leadership	Program director (0.25 FTE), 4 principal investigators (0.1 FTE each)
Assets	Freezers, equipment
Consumables	N/A
Information Technology	7 IT staff, including product managers, developers, and QA specialists, manage multiple biobanking software applications. The key systems relate to biospecimen management/processing, recruitment, and biospecimen/data queries and distribution

**Table 4 jpm-12-01323-t004:** Internal metrics.

Participants	135,000
Number of aliquots	1,244,395
Extent of data held	Genotyping data for ~65,000 participants and whole exome sequencing data for ~54,000 participants. The entire electronic health record for all participants is also available.
Biobank certification/ accreditation	CLIA certified
Biobanking research grants	None
Publications on internal biobank activities	2

**Table 5 jpm-12-01323-t005:** Biobank outputs.

Inquiries managed	The Biobank has completed 1142 distributions of biospecimens/data to 446 studies. From 15 November 2018 to 6 June 2022, the Biobank fielded 2358 inquiries from investigators.
Distributed biospecimens	152,650
Utilization rate of biospecimens	13% of total biospecimens collected have been distributed to date.
Research grants supported	Studies that have received Biobank services received research grants representing ~$511,000,000 in funding.
Research projects supported	The Biobank has distributed biospecimens/data to 446 unique studies. Of note, the Biobank collaborated with several studies focused on COVID-19 collections in 2020–2022, building up a repository of COVID-19 biospecimens.
Cost recovery	The cost of biospecimens/data covers a part of the Biobank’s cost model. All the data distributions are free of charge.
Research collaborations	33
Clinical practice changes	Introduction of return of genomic results into the clinical workflow.
Publications on biobanking and biobank outputs	5
Patents	0

**Table 6 jpm-12-01323-t006:** Demographics of the Mass General Brigham Hospitals and Community Health Centers.

Race	Biobank Participants: Main Hospitals	Biobank Participants: Community Health Centers
American Indian/Alaska Native	0.1%	0.3%
Asian	2.6%	1.2%
Black	5.2%	2.7%
Hispanic	5.1%	80.8%
Other	1.0%	1.4%
Two or More	0.6%	0.2%
Unknown	2.5%	0.5%
White	82.8%	12.9%

**Table 7 jpm-12-01323-t007:** Lessons Learned in the Development and Operation of the MGB Biobank.

Issue	Lessons Learned
Participant recruitment	• In-person recruitment in clinic-based settings is effective. • Remote electronic enrollment provides a low-cost recruitment mechanism but may limit diversity. • Integration with clinical workflows can drive low-cost, effective recruitment mechanisms.
Biospecimen collection	• Integration with clinical workflows is essential to efficient biospecimen collection.
IT infrastructure	• Leveraging clinical systems whenever possible to reduce costs and create a more seamless experience for patients and participants. • Flexibility, as always, is key when operating within a large healthcare system with constantly evolving clinical systems and structures.
Human subjects compliance	• Close collaboration with IRB throughout design and operations is helpful to pre-empt and address human subjects’ issues.
Return of genetic results	• Need for a smooth transition from research return of results to clinical care. • A substantial fraction of participants with actionable results do not pursue disclosure.
Diversity and equity	• Engagement with diverse communities and recruitment in clinical sites serving diverse patient populations is essential.
Sustainability	• Trade-off between facilitating broad use by the research community and cost recovery.

## Data Availability

A portion of the MGB Biobank genomic data are available in dbGAP as part of the eMERGE consortium phase 3 (https://www.ncbi.nlm.nih.gov/projects/gap/cgi-bin/study.cgi?study_id=phs001584.v2.p2 accessed on 15 November 2021) and as part of the Center Common Disease Genomics (https://www.ncbi.nlm.nih.gov/projects/gap/cgi-bin/study.cgi?study_id=phs002018.v1.p1 accessed on 15 November 2021). Additional MGB Biobank data are not currently publicly available due to restrictions on the data.

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
