# Peer review of "The Evolution of a Large Biobank at Mass General Brigham"

_jpm, 2022, doi:10.3390/jpm12081323_

Round 1

Reviewer 1 Report

I have enjoyed this manuscript very much. Essentially it is about what the title says: the evolution of a large Biobank in the United States. It provides a very detailed overview of all the many angles that affect the design and activities of a large multi-centric Biobank. The authors discuss a lot of the challenges that the Biobank faces, many of the changes implemented in response to new situations, and to what extent they are useful to improve the activities and goals of the Biobank. It is an easy read, the flow of sections is well designed, and the tables, figures and references are all useful. Of note, the authors provide discussion of the shortcomings and problems associated with some of the changes implemented in the procedures by the Biobank. This frank evaluation of the pros and cons is particularly interesting. 

In summary, the manuscript is very insightful and very well written. If anything, after reading it, I feel a bit curious and eager for more details regarding the role of leadership and the institutional support that would maximize the impact of the Biobank. Lessons in that regard would be useful for the design of new Biobanks, and finding strategies to maximize the outcomes depending on the institutional actors involved. But rather than a formal request, this is more of a general comment and I would like to leave it entirely to the authors’ discretion.

Author Response

We thank the reviewer for their positive comments on our manuscript.  We appreciate the suggestions to consider further elaboration about the role of leadership and institutional support.  The question of what institutional support would maximize the Biobank's future would have to be framed in terms of what support is realistic and what institutional circumstances (many specific to MGB) would need to be considered--e.g. support for transport and pathology processing at community hospitals.  Given the length of the manuscript at present, we've elected not to add further discussion at this point. 

Reviewer 2 Report

1.     Line 127; When referring to the 1.2 million contacted patients, is this 1.2 million unique patients or were some patients contacted multiple times?

2.     Table 1: The percentage reported in Table 1 would be more interpretable if the Diversity and Equity were discussed here and the percentages presented in Table 6 Demographics if the Mass General Brigham Hospitals and Community Health Centers to provide context of the patients population. 

3.     Lines 140-143: The definition of ‘Collaborating Studies’ is unclear. Here 33 studies are mentioned but on line 73-74 & 379 genomic data has been distributed to 247 studies while biospecimens have been distributed to 208 studies (line 378). Table 6 states 446 unique studies have received biospecimens/data. A clear accounting would be nice.

4.     Table 2: The caption should explain what Yes and No represent in the Fact Sheet Topic. That is, I am interpreting that there is no information provided about Bruising or infection from blood draw, but I don’t see how that could not possibly be included in the Informed Consent process.

5.     Lines 288-289; 315-320: Regarding the query tool for investigators to identify participants, are results returned in a de-identified manner? That is, is this IT system set up using some type of honest broker system. Do investigators accessing these data require IRB approve? Any specific human subjects training?

6.     Line 322-323: It is unclear what is meant by, “The Charlson Comorbidity Index is used to calculate healthy controls.” Does this mean Charlson Comorbidity Index is calculated for healthy controls? How are healthy controls identifiable?

7.     Figure 3: Is the database linked to administrative data such as the Social Security Death Index for tracking patient survival?

8.     Lines 345-347 & 367-369: Given that the Biobank distributes biospecimens to MGB investigators, is there an expectation that assay results performed on the biospecimens will be entered into the Biobank database? It is unclear how the section starting on line 451 Return of Research Results can be effective without requiring investigators to populated the Biobank database.

9.     Lines 430-434 or 526-530: Is there a specific requirement to acknowledge the Biobank in publications? If so, is that being tracked?

10.  Table 6: What is meant by Utilization rate of biospecimens? How many research grants is the Biobank supporting (in addition to total dollars funded)?

11.  In the current version of the manuscript there are 2 tables named Table 6.
